# Tissue-Specific Regulation of HNK-1 Biosynthesis by Bisecting GlcNAc

**DOI:** 10.3390/molecules26175176

**Published:** 2021-08-26

**Authors:** Haruka Kawade, Jyoji Morise, Sushil K. Mishra, Shuta Tsujioka, Shogo Oka, Yasuhiko Kizuka

**Affiliations:** 1Graduate School of Natural Science and Technology, Gifu University, Gifu 501-1193, Japan; a4521031@edu.gifu-u.ac.jp; 2Department of Biological Chemistry, Human Health Sciences, Graduate School of Medicine, Kyoto University, Kyoto 606-8501, Japan; morise.jyoji.2n@kyoto-u.ac.jp (J.M.); okaken.tsujioka@gmail.com (S.T.); oka.shogo.7w@kyoto-u.ac.jp (S.O.); 3Glycoscience Center of Research Excellence, The University of Mississippi, Oxford, MS 38677, USA; sushil@olemiss.edu; 4Institute for Glyco-core Research (iGCORE), Gifu University, Gifu 501-1193, Japan

**Keywords:** bisecting GlcNAc, glucuronyltransferase P (GlcAT-P), glucuronyltransferase S (GlcAT-S), glycosylation, *N*-acetylglucosaminyltransferase III (GnT-III), human natural killer—1 (HNK-1)

## Abstract

Human natural killer—1 (HNK-1) is a sulfated glyco-epitope regulating cell adhesion and synaptic functions. HNK-1 and its non-sulfated forms, which are specifically expressed in the brain and the kidney, respectively, are distinctly biosynthesized by two homologous glycosyltransferases: GlcAT-P in the brain and GlcAT-S in the kidney. However, it is largely unclear how the activity of these isozymes is regulated in vivo. We recently found that bisecting GlcNAc, a branching sugar in *N*-glycan, suppresses both GlcAT-P activity and HNK-1 expression in the brain. Here, we observed that the expression of non-sulfated HNK-1 in the kidney is unexpectedly unaltered in mutant mice lacking bisecting GlcNAc. This suggests that the biosynthesis of HNK-1 in the brain and the kidney are differentially regulated by bisecting GlcNAc. Mechanistically, in vitro activity assays demonstrated that bisecting GlcNAc inhibits the activity of GlcAT-P but not that of GlcAT-S. Furthermore, molecular dynamics simulation showed that GlcAT-P binds poorly to bisected *N*-glycan substrates, whereas GlcAT-S binds similarly to bisected and non-bisected *N*-glycans. These findings revealed the difference of the highly homologous isozymes for HNK-1 synthesis, highlighting the novel mechanism of the tissue-specific regulation of HNK-1 synthesis by bisecting GlcNAc.

## 1. Introduction

Protein functions are regulated by post-translational modifications, including glycosylation [1]. Glycosylation, the most common form of post-translational modification, has significant effects on protein behavior, such as interaction with other biomolecules, subcellular localization, stability, and activity [1,2], thereby maintaining the functions of proteins and cells. Conversely, dysregulated expression of glycans leads to the development of various diseases, including cancer, diabetes, and Alzheimer’s disease (AD) [3,4,5]. Therefore, elucidation of the mechanisms of glycan expression is pivotal to understanding how protein functions are regulated under physiological and pathological conditions.

Among several types of glycans on proteins, *N*-glycosylation is highly conserved in eukaryotes [6]. In the endoplasmic reticulum (ER), the common *N*-glycan precursor is transferred to a polypeptide [7], and the transferred *N*-glycan is then trimmed by glucosidases and mannosidases. Thereafter, *N*-glycosylated proteins are transported to the Golgi apparatus, where *N*-glycans are further modified to form a wide variety of structures by the concerted and competitive actions of many Golgi-resident glycosyltransferases [2,8]. A certain *N*-glycan structure is frequently expressed in a protein-, cell-, and tissue-specific manner, but little is known about the mechanisms by which this is achieved. To elucidate the complexity of *N*-glycan expression, it is crucial to understand the mechanism regulating the in vivo activity of each glycosyltransferase involved in *N*-glycan biosynthesis.

Bisecting GlcNAc, the central GlcNAc branch linked to β-mannose synthesized by GnT-III (also designated as MGAT3) [9] (Figure 1A), has been shown to significantly impact the biosynthesis of entire *N*-glycan structures. For example, once *N*-glycan is modified with bisecting GlcNAc, the bisected glycan is no longer a substrate for other branching enzymes, such as GnT-IV and GnT-V [10,11]. Moreover, the presence of bisecting GlcNAc was also shown to inhibit the synthesis of the alpha-Gal epitope in *N*-glycans [12]. These findings suggest that an *N*-glycan profile in a certain protein or cell is significantly affected by the level of bisecting GlcNAc.

The expression of bisecting GlcNAc and GnT-III is tissue-specific, with the highest expression in the brain and the kidney [13,14]. GnT-III KO mouse studies have revealed that bisecting GlcNAc regulates the functions of the modified proteins in the brain, resulting in the promotion of AD pathology [15,16]. BACE1 (beta-site APP cleaving enzyme-1), a protease responsible for the production of amyloid-beta (Aβ) in AD, is heavily modified with bisecting GlcNAc [16,17], and BACE1 without bisecting GlcNAc is relocated to lysosomal compartments, leading to improvement of the AD pathology in the brains of GnT-III KO mice [16]. Furthermore, loss or overexpression of GnT-III in cancer cells was also reported to greatly affect tumor growth and malignancy [18,19]. These findings suggest that glycan modification with bisecting GlcNAc plays a significant role in the regulation of glycoprotein functions and pathogenesis. More recently, our *N*-glycomic study of the brains of GnT-III KO mice revealed that bisecting GlcNAc also suppresses the biosynthesis of many *N*-glycan terminal epitopes, including Lewis-type fucose, sialic acid, and the HNK-1 epitope [20]. Our enzyme assays and molecular dynamics (MD) simulation further revealed that the presence of bisecting GlcNAc inhibits the activity of many glycosyltransferases by altering *N*-glycan conformation. This suggests that bisecting GlcNAc has broad suppressive effects on *N*-glycan modifications.

HNK-1, a terminal glycan structure in *N*- and *O*-mannose glycans predominantly expressed in the brain [21,22], comprises a 3-sulfated glucuronic acid (GlcA) linked to galactose via a β1,3-linkage. Two glucuronyltransferases (GlcAT-P and GlcAT-S, also designated as B3GAT1 and B3GAT2, respectively) and a sulfotransferase (HNK-1ST, also designated as CHST10) have been identified as biosynthetic enzymes [23,24,25] (Figure 1A). Although the functional differences between GlcAT-P and GlcAT-S have not been fully clarified, these two isozymes were found to play distinct roles in vivo. Knocking out GlcAT-P resulted in almost complete loss of HNK-1 expression in the brain, leading to impaired memory and learning functions [26]. By contrast, GlcAT-S, but not GlcAT-P and HNK-1ST, was shown to be selectively expressed in mouse kidney and to biosynthesize non-sulfated HNK-1 epitope in the kidney [27,28] (Figure 1A). These findings indicate that HNK-1 (which hereafter includes its non-sulfated form) synthesis is regulated in a tissue-specific manner, and GlcAT-P is dominant in the brain whereas GlcAT-S is so in the kidney. Considering that GnT-III is also highly expressed in the brain and the kidney and that bisecting GlcNAc inhibits the biosynthesis of HNK-1 in the brain, we hypothesized that the biosynthesis of non-sulfated HNK-1 by GlcAT-S in the kidney could also be negatively regulated by bisecting GlcNAc.

In this study, to understand the regulatory mechanism of GlcAT-S by bisecting GlcNAc, we compared the expression levels of HNK-1 in the kidneys of WT and GnT-III KO mice. We also conducted in vitro activity assays of GlcAT-P and GlcAT-S toward bisected and non-bisected acceptor glycans. Our results revealed that the effects of bisecting GlcNAc on the activity of GlcAT-P and GlcAT-S are completely different. Our findings provide new insights into the regulatory mechanisms of HNK-1 expression and the physiological functions of bisecting GlcNAc.

## 2. Results

### 2.1. HNK-1 Expression Is Inhibited by Bisecting GlcNAc in Mouse Brain but Not in Kidney

To investigate whether bisecting GlcNAc also affects the biosynthesis of HNK-1 in the kidney, we first compared the expression levels of HNK-1 glycan between wild-type (WT) and GnT-III KO (Mgat3^−/−^) mice in both the brain and the kidney by Western blotting. Our previous glycomic study using LC-MS demonstrated that knocking out GnT-III in mice caused loss of bisecting GlcNAc from *N*-glycans in both the brain and the kidney [20]. Consistent with this, GnT-III KO brain and kidney both showed decreased reactivity with E4-PHA lectin (Figure 1B,C), which preferentially recognizes bisected *N*-glycans [29]. In the brain, several glycoproteins were selectively modified with HNK-1 glycan by GlcAT-P (Figure 1B) [26,30], and the level of HNK-1 expression was increased in the brains of GnT-III KO mice compared with that in those of WT mice (Figure 1B), as reported previously [20]. In mouse kidney, the limited membrane proteins, including meprin alpha and aminopeptidase N, are modified with non-sulfated HNK-1 glycan, which is specifically recognized by M6749 mAb (Figure 1D) [27]. We confirmed that the biosynthesis of non-sulfated HNK-1 in mouse kidney almost completely depends on GlcAT-S (Figure 1D). In sharp contrast to the increased expression of HNK-1 in GnT-III KO brain, the level of non-sulfated HNK-1 in the kidneys of GnT-III KO mice was comparable to that in the WT (Figure 1D). We also immunostained kidney sections from WT and GnT-III KO mice with M6749 mAb. Non-sulfated HNK-1 is expressed in the apical membrane of proximal tubules in the cortex [27], and again this staining disappeared in GlcAT-S KO (B3gat2^−/−^) mice (Figure 1E). Similar to the results of Western blotting, the pattern and level of M6749 staining in the kidneys of GnT-III KO mice were similar to those in WT (Figure 1E). Together, these results showed that the biosynthesis of HNK-1 is upregulated by GnT-III deficiency in the brain but not in the kidney.

### 2.2. GlcAT-S Is Fully Active toward N-glycans with Bisecting GlcNAc

The above findings demonstrated that the levels of GlcAT-P products in the brain, but not those of GlcAT-S products in the kidney, were increased in the absence of bisecting GlcNAc. On the basis of this, we hypothesized that GlcAT-P and GlcAT-S have different preferences for bisected *N*-glycans in their catalytic reactions, even though they are highly homologous (49.5% identity for rat enzymes).

To test this possibility, we aimed to directly measure the activity of these enzymes toward bisected and non-bisected *N*-glycan substrates in vitro. For this purpose, we purified recombinant soluble proteinA-tagged GlcAT-P and GlcAT-S from COS7 culture media (Figure 2A). As acceptor glycans, fluorescently labeled galactose-terminated biantennary *N*-glycans with or without bisecting GlcNAc (bisectGGnGGnbi-PA and GGnGGnbi-PA [20]) were prepared. The purified enzymes were incubated with either substrate, and the reaction mixtures were analyzed by HPLC to separate the acceptor and the products (Figure 2B). Consistent with the previous study, GlcAT-P exhibited weaker activity toward the bisected-type substrate than toward its non-bisected counterpart (Figure 2C, upper) [20]. By contrast, GlcAT-S showed almost equivalent activity toward bisected and non-bisected substrates (Figure 2C, lower). The overall activity of GlcAT-P was higher than GlcAT-S, consistent with the previous reports [31,32]. These findings revealed that the presence of bisecting GlcNAc in acceptor glycan inhibits the activity of GlcAT-P but not that of GlcAT-S, which could cause the different effects of GnT-III deficiency on the expression of HNK-1 between brain and kidney.

### 2.3. Different Effects of Bisecting GlcNAc on Intracellular Activity of GlcAT-P and GlcAT-S

We next investigated whether the inhibition of GlcAT-P, but not GlcAT-S by bisecting GlcNAc, is also observed in cells. We first overexpressed myc-tagged GnT-III with FLAG-tagged GlcAT-P or GlcAT-S in HEK293 cells. As GlcATs and HNK-1 glycan are not endogenously expressed in HEK293 cells, exogenous expression of GlcAT-P or -S led to the production of M6749-reactive HNK-1 glycans on various cellular proteins (Figure 3A). The levels of GlcAT-P-produced HNK-1 glycans were largely suppressed by GnT-III overexpression (Figure 3A, 2nd and 5th lanes), and the suppressive effects of GnT-III overexpression were also observed for GlcAT-S (Figure 3A, 3rd and 6th lanes). This shows that the overexpression of GnT-III at high levels suppresses both GlcAT-P and GlcAT-S in cells by unknown mechanisms. Although the presence of bisecting GlcNAc suppressed the activity of GlcAT-P but not that of GlcAT-S in the enzyme assay (Figure 2), overexpression of GnT-III in cells might affect the activity or localization of these enzymes as both GlcAT-P and -S are *N*-glycosylated.

We then examined the effects of deletion of endogenous GnT-III on HNK-1 biosynthesis in cultured cells. To this end, GnT-III KO Neuro2A cells were generated by CRISPR-Cas9. Genotyping PCR (Figure 3B) and GnT-III activity assay using the cell lysates (Figure 3C) confirmed GnT-III deficiency in the established KO clone. FLAG-tagged GlcAT-P or -S was expressed in WT or GnT-III KO Neuro2A cells, and their HNK-1-producing activity was examined by Western blotting with M6749 mAb. As Neuro2A cells do not express GlcATs endogenously, exogenous expression of GlcATs resulted in the expression of M6749-reactive HNK-1 glycans on various proteins (Figure 3D). The results showed that the intracellular activity of GlcAT-P was higher in Neuro2A GnT-III KO cells than in WT cells, but such an increase in the HNK-1 level in KO cells was not observed for GlcAT-S (Figure 3D). These findings indicate that the biosynthetic activities of GlcAT-P and GlcAT-S are also distinctly regulated by bisecting GlcNAc in cultured cells.

### 2.4. Molecular Dynamics (MD) Simulations of the Binding between GlcATs and N-glycans with or without Bisecting GlcNAc

To obtain structural insights into how the two GlcATs show different preferences for bisected *N*-glycans, the structures of GlcAT-P and GlcAT-S in complex with bisected or non-bisected *N*-glycan were modeled based on the crystal structures of these enzymes (Figure 4A–H) [33,34]. For each enzyme-glycan pair, two models were constructed in which either the α(1,3)-arm or the α(1,6)-arm was bound to the catalytic pocket.

The MD simulation and clustering of glycan conformations that were obtained from MD showed that the different preferences of GlcAT-P and GlcAT-S toward bisectGGnGGnbi can be explained by the binding energies of bisected *N*-glycans to GlcAT-P and GlcAT-S (Figure 4I and Appendix A). The GGnGGnbi binds more strongly to GlcAT-P than to GlcAT-S, whereas bisectGGnGGnbi binds well to GlcAT-S but poorly to GlcAT-P. This agrees with the activity assays of GlcAT-P and GlcAT-S showing that GlcAT-P prefers GGnGGnbi, whereas GlcAT-S has almost equivalent activity toward GGnGGnbi and bisectGGnGGnbi (Figure 2C). In particular, GlcAT-P did not bind to bisectGGnGGnbi in α(1,6)-binding mode at all, and the glycan drifted away from the reactive site quickly during MD (Figure 4D). The weak and/or lack of binding of bisectGGnGGnbi is due to a different conformation of the R142–I155 loop in GlcAT-P from that of GlcAT-S, which occupies the space where bisecting GlcNAc should fit (Figure 4J), causing steric clashes with bisectGGnGGnbi. The α(1,3)-binding mode of GGnGGnbi in both GlcAT-P and GlcAT-S predominantly exists in back-fold conformation, where the α(1,6)-branch folds back and stacks up over the chitobiose core.

Taken together, these structural insights further showed that bisecting GlcNAc weakens the affinity to GlcAT-P but not GlcAT-S, allowing the enzyme-specific regulation of HNK-1 synthesis by bisecting GlcNAc.

## 3. Discussion

In this study, we showed that the levels of GlcAT-P products in the brain, but not the levels of GlcAT-S products in the kidney, were increased in the absence of bisecting GlcNAc. This can be explained by the different preferences of GlcAT-P and GlcAT-S to bisected *N*-glycans (Figure 2 and Figure 4). Although the specific activity of GlcAT-P is totally higher than GlcAT-S [31,32], HNK-1 expression in brain and kidney exclusively depends on GlcAT-P and GlcAT-S, respectively. Therefore, our findings suggest that the tissue-specific expression of the two isozymes (GlcAT-P and -S) with distinct tolerances for bisecting GlcNAc allows the tissue-specific regulation of HNK-1 synthesis by bisecting GlcNAc.

Although the overall structures of GlcAT-P and GlcAT-S are highly conserved [33,34], a few structural differences could lead to different specificities of these two enzymes. As shown in Figure 4J, the R142–I155 loop in GlcAT-P could cause a steric clash with bisecting GlcNAc. Although the MD simulation showed that a bisected glycan is not a good binder for GlcAT-P and a good binder for GlcAT-S, our enzyme assays showed that GlcAT-P has higher activity than GlcAT-S even for bisected glycans. The reason for this apparent discrepancy cannot be explained by our MD simulation at present. Furthermore, our previous study revealed that the replacement of the key residues (Trp234 and Ala309 in GlcAT-S, corresponding to Phe245 and Val320 in GlcAT-P) in the acceptor sugar-binding site leads to the strict specificity of GlcAT-P toward LacNAc (Galβ1-4GlcNAc) and tolerance of GlcAT-S for both LacNAc and Lacto-N-biose (Galβ1-3GlcNAc) [34].

The different specificities of the two HNK-1-synthesizing isozymes toward bisected glycans could provide insights into why these two similar enzymes are present. As the levels of bisecting GlcNAc, which is also highly expressed in brain and kidney [13], are dynamically altered in response to various stimuli and stresses [35,36], this could in turn affect the expression levels of HNK-1 glycans, particularly in the brain. Although the functions of non-sulfated HNK-1 in the kidney synthesized by GlcAT-S remain unresolved [27,28], fluctuation of the level of GlcAT-S-synthesized non-sulfated HNK-1 may be undesirable in the kidney. By contrast, in the brain, the negative correlation between the levels of GlcAT-P-synthesized HNK-1 and bisecting GlcNAc may be required for physiological functions. Both glycan epitopes are highly expressed in neurons [14,22,37], and we previously showed that HNK-1 is required for morphological maturation and plasticity of synapses [37,38]. As the expression level of HNK-1 is elevated in the GnT-III KO brain [20], analysis of the functions and morphology of GnT-III KO neurons could elucidate the roles of suppression of HNK-1 expression by bisecting GlcNAc. Additionally, the suppression of HNK-1 expression by bisecting GlcNAc could be related to brain diseases. The increased expression of bisecting GlcNAc and the decreased expression of HNK-1 have been suggested to be involved in the pathology of AD [16,39,40]. Studies have reported that HNK-1 is selectively expressed on the major subunit of AMPA-type glutamate receptor GluA2 to maintain synaptic functions [37,38], and GlcAT-P KO (*B3gat1*^−/−^) mice displayed impaired learning and memory [26]. On the basis of these findings, it is possible that insufficient HNK-1 synthesis by aberrantly upregulated GnT-III expression is related to the development or aggravation of neural disorders, including AD.

As to the mechanism explaining why the level of HNK-1 is increased in the brain but not in the kidney in the absence of bisecting GlcNAc, we focused on the acceptor glycan specificity of the enzymes that synthesize HNK-1. Alternatively, the polypeptide part of the HNK-1-carrying proteins may also influence the levels of HNK-1. We previously reported that GlcAT-P showed higher activity toward a glycoprotein substrate than toward a disaccharide substrate (Gal-GlcNAc) in vitro [31,32]. In vivo, the carrier proteins of HNK-1 are highly selective and completely different between brain and kidney. In the brain, limited cell adhesion molecules and the neurotransmitter receptors are modified by GlcAT-P, including neural cell adhesion molecule (NCAM), myelin-associated glycoprotein (MAG), and GluA2 [21,22]. By contrast, in the kidney, the metalloproteases in the apical membrane and an extracellular matrix molecule are specifically modified by GlcAT-S, including meprin alpha, aminopeptidase N, and laminin111 [27,28]. Because the activity of GlcAT-P and GlcAT-S for each carrier glycoprotein is unclear, investigation of their activity for various glycoproteins will be important to elucidate the detailed mechanisms by which HNK-1 expression is regulated in vivo.

Previous studies have shown that various enzymes for branching and terminal modifications of *N*-glycans are inhibited by bisecting GlcNAc, including GlcAT-P [10,20]. Here, we showed that GlcAT-S, despite being highly homologous to GlcAT-P, is not inhibited by bisecting GlcNAc. A similar finding was made for fucosyltransferases, and we reported that FUT2, 4, 7, and 9 are inhibited by bisecting GlcNAc, while FUT1 is not [20]. These findings suggest that many enzymes acting at the mid to late stages of *N*-glycan biosynthesis are commonly inhibited by bisecting GlcNAc, whereas some enzymes do not distinguish the core structures of *N*-glycans, including bisecting GlcNAc, but recognize only terminal acceptor sugar residues. Structural studies of these enzymes in complex with various types of acceptor *N*-glycans or glycoproteins should provide further information regarding how glycosyltransferases distinguish the core structures of *N*-glycans.

In conclusion, our study highlights a mechanism of tissue-specific regulation of a specific glyco-epitope, HNK-1. This regulation is mediated by another type of glycan epitope, namely, bisecting GlcNAc. Similarly, several recent reports demonstrated that a certain type of glycan regulates the expression of other classes of glycans. For instance, knockout of mannosidases involved in *N*-glycan synthesis leads to unexpected elevation of the enzymes responsible for glycolipid and hyaluronic acid synthesis [41]. Additionally, deletion of one *N*-glycan branch was reported to evoke LacNAc extension in another branch [42]. These lines of accumulating evidence suggest that mammalian cells and tissues have various self-correcting mechanisms to maintain the levels and functions of glycans and that bisecting GlcNAc plays a central role in fine-tuning the complex structures of *N*-glycans. Investigation of the detailed specificity and the structures of other glycosyltransferases acting on *N*-glycans will provide further information about the complex regulatory mechanisms of *N*-glycan biosynthesis.

## 4. Materials and Methods

### 4.1. Reagents

The following antibodies and a lectin were used: mouse anti-HNK-1 (BD Bioscience, Tokyo, Japan, 559048), M6749 [43], mouse anti-VDAC1 (Abcam, ab14734, Waltham, MA, USA), mouse anti-FLAG (Sigma, Tokyo, Japan F1804), mouse anti-GAPDH (Millipore, Darmstadt, Germany MAB374), HRP-conjugated anti-mouse IgG (GE Healthcare, NA934V), HRP-conjugated anti-mouse IgM (Biosource, CA, USA), and E4-PHA (J-CHEMICAL, Tokyo, Japan J111). For labeling E4-PHA with HRP, lyophilized E4-PHA was diluted with PBS to a concentration of 2 mg/mL and labeled with HRP using Peroxidase Labeling Kit-NH_2_ (Dojindo, Kumamoto, Japan), in accordance with the manufacturer’s protocol.

### 4.2. Animal Experiments

All mice were from a C57BL/6 genetic background. The generation of GnT-III (*Mgat3*)-deficient mice was described previously [44] and they were generously provided by Dr. Jamey D. Marth (University of California-Santa Barbara). GlcAT-S (*B3gat2*)-deficient mice were generated as described previously [30]. The mice were housed (four or fewer per cage) at 23 ± 3 °C and 55 ± 15% humidity. The lighting conditions were 12 h light:12 h dark.

### 4.3. Preparation of Membrane and Soluble Fractions

Mouse brains and kidneys were homogenized in eight volumes of homogenization buffer (50 mM Tris-HCl, pH 7.4, 0.15 M NaCl, protease inhibitor cocktail) using a Potter grinder. The homogenate was centrifuged at 1000× *g* for 10 min at 4 °C to remove nuclei and unbroken cells. The supernatant was ultracentrifuged at 105,000× *g* for 30 min at 4 °C. The resultant pellet and supernatant were used as the membrane and soluble fractions, respectively, and subjected to Western and lectin blotting.

### 4.4. Western and Lectin Blotting, and CBB Staining

Proteins were resolved by 5–20% SDS-PAGE. For CBB staining, proteins separated in the gel were stained with GelCode Blue Safe Protein Stain (Thermo Fisher Scientific) and visualized using FUSION-SOLO 7s EDGE (Vilber Lourmat, Collegien, France). For Western blotting, proteins separated in the gel were transferred to nitrocellulose membranes. The membranes were blocked with TBS containing 5% skim milk and 0.1% Tween 20 and were incubated with primary antibody, followed by incubation with HRP-conjugated secondary antibody. For lectin blotting, the membranes were blocked with TBS containing 1% BSA and 0.1% Tween 20 and incubated with HRP-labeled lectin diluted in TBS containing 1% BSA and 0.1% Tween 20. Protein bands were detected with Western Lightning Plus-ECL (PerkinElmer Life Sciences, WF, USA) using FUSION-SOLO 7s EDGE (Vilber Lourmat, Collegien, France).

### 4.5. Immunofluorescence Staining

Deeply anesthetized mice were transcardially perfused with PBS and then with PBS containing 4% paraformaldehyde. Tissues were cryoprotected with 30% sucrose and cut into 40 μm thick slices using a microtome. The floating sections were incubated with M6749 monoclonal antibody (1:400) diluted in 3% BSA/PBS for 1 h. After washing with PBS containing 0.1% TritonX-100, those sections were incubated with TRITC-anti-mouse IgM antibody (Cappel, 1:400) diluted in 3% BSA/PBS for 1 h. To visualize the nuclei, after washing with PBS containing 0.1% TritonX-100, the sections were incubated with DAPI solution (Nacalai Tesque, 1:1000) diluted in 3% BSA/PBS for 5 min. All procedures were performed at room temperature. The images were obtained using a Fluoview laser confocal microscope system (FV1000-D IX81, Olympus, Tokyo, Japan).

### 4.6. Cell Culture and Transfection

HEK293, Neuro2A, Neuro2A/GnT-III KO, and COS7 cells were cultured in Dulbecco’s modified Eagle’s medium containing 10% fetal bovine serum and 50 μg/mL kanamycin under 5% CO_2_ at 37 °C. For cellular experiments, plasmids were transfected into cells plated on a 10 or 6 cm dish at 70–80% confluence using Lipofectamine 3000 transfection reagent (Thermo Fisher Scientific), in accordance with the manufacturer’s protocol. For the expression and purification of recombinant proteinA-tagged GlcAT-P and -S, plasmids were transfected into COS7 cells using polyethyleneimine MAX (Polyscience, WG, USA).

### 4.7. Plasmid Construction

The plasmids for proteinA-GlcAT-P (pEF-protA-BOS/rat GlcAT-P) and proteinA-GlcAT-S (pEF-protA-BOS/rat GlcAT-S) were constructed as described previously [31]. The plasmids for N-terminally 3xFLAG-tagged GlcAT-P (p3xFLAG-CMV10/rat GlcAT-P) and GlcAT-S (p3xFLAG-CMV10/rat GlcAT-S) were also generated as described previously [45]. Additionally, the plasmid for C-terminally myc-His-tagged GnT-III (pcDNA6-myc HisA/human GnT-III) was constructed as described previously [20]. For knocking out the *Mgat3* gene in Neuro2A cells, two px330-puro plasmids having either one of two guide sequences (ATGACGTCTTTATCATCGAC and AGGGCATCTACTTTAAACTC) inserted into the BbsI site were used.

### 4.8. Purification of Recombinant Enzyme

For purification of recombinant soluble GlcAT-P and GlcAT-S, the expression plasmids were transfected into COS7 cells on 15 cm dishes at 70–80% confluence using polyethyleneimine MAX. After 6 h, the medium was replaced with Opti-MEM I, followed by further incubation for 72 h. The proteinA-tagged GlcAT-P and GlcAT-S were purified from the medium using an IgG column (IgG Sepharose 6 Fast Flow, Cytiva, Tokyo, Japan).

### 4.9. Glycosyltransferase Activity Assay

The activity of GlcAT-P and -S was measured as described previously [20] with minor modifications. In brief, purified GlcAT-P or GlcAT-S was incubated in 10 μL of the reaction buffer [125 mM MES pH 6.2, 10 mM MnCl_2_, 0.2 M GlcNAc, 0.5% Triton X-100, 1 mg/mL BSA, 1 mM UDP-GlcA, and 2.5 μM fluorescently labeled acceptor substrate (GGnGGnbi-PA or bisectGGnGGnbi-PA)] at 37 °C for 1 h (GlcAT-P) or 3 h (GlcAT-S). The acceptor substrates were prepared by transferring galactose and bisecting GlcNAc to GnGnbi-PA by recombinant B4GALT1 and GnT-III, as described previously [20]. The glucuronyltransferase reaction was stopped by boiling at 95 °C for 2 min and 40 μL of water was added to the mixture. After centrifugation at 15,000× *g* for 3 min, 10 μL of the supernatant was injected into an HPLC system equipped with an ODS column (Tosoh; 4.6 × 150 nm) and analyzed in isocratic mode (20 mM ammonium acetate, pH 4.0, containing 0.2% 1-butanol). GnT-III activity toward fluorescently labeled biantennary glycan (GnGnbi-PA) was analyzed as described previously [46].

### 4.10. Generation of GnT-III KO Neuro2A Cells

Neuro2A cells were co-transfected with two px330-puro plasmids both targeting *Mgat3* genes. After 24 h, puromycin was added to the medium to a final concentration of 4 μg/mL, followed by incubation for 2 days. The surviving cells were subjected to limiting dilution, and the genotype of each clone was checked by PCR using the following primers: AGTACATCCGCCACAAGGTG and CAAGTAGCGGAACTGGTCAT.

### 4.11. Building GlcAT/N-Glycan Complex Models

Structures of GlcAT-P (UniProt ID: O35789) and GlcAT-S (UniProt ID: Q9Z137) are shown in Prime (Schrödinger, LCC, NY 2020). The GlcAT-P sequence started from Asp75, whereas GlcAT-S started from Asp24. A search for their sequential homologs was performed in the Protein Databank (PDB) using BLAST, and the structures of human GlcAT-P and GlcAT-S were found to have more than 91% sequence identity. The lowest-resolution structures of each of PDB IDs 1V84 [33] and 2D0J [34] were used to model structures of murine GlcAT-P and GlcAT-S, respectively. The homology models were built using default parameters in the knowledge-based model building approach of Prime. Subsequently, the structures of GlcAT-P and GlcAT-S complexes with biantennary (GGnGGnbi) and bisected biantennary (bisectGGnGGnbi) glycans were modeled by pair-fitting *N*-glycan branches to the Galβ(1–4)GlcNAc fragment in the crystal structure of the human GlcAT-P α1–3-binding mode. We extracted the starting structure of GGnGGnbi from the *N*-glycosylation site of the human ST6GAL1 crystal structure (PDB ID: 4JS1) [47]. In contrast, the structure of bisectGGnGGnbi was created manually by adding bisecting GlcNAc to GGnGGnbi. The modeled complex looked similar to that shown in our earlier study and also in Figure 2C of the paper by Kuhn et al. [47]. A total of four complexes were prepared for each case, where (i) the α1–3 branch of GGnGGnbi was bound to the catalytic site; (ii) the α1–6 branch of GGnGGnbi was bound to the catalytic site; and (iii) and (iv) are the same binding modes for bisectGGnGGnbi where an additional GlcNAc is attached to GlcNAc-2. These complexes were further subjected to classical MD simulations and binding free-energy calculations.

### 4.12. Molecular Dynamics and Binding Energy Calculation

All eight complexes were solvated in an octahedral TIP3P water box extending 12 Å from each side of the complex. Amber force field ff14SB was used to treat the protein, whereas GLYCAM06 (version j-1) was used for glycans. The Li/Merz ion parameters were used for the Mn^2+^ ion in the GlcAT-P active site [48]. A multi-step protocol published elsewhere [49] was used to equilibrate the complexes. Finally, 100 ns MD of each complex was performed at NPT, using the following MD settings: temperature of 300 K, temperature scaling by Langevin dynamics (collision frequency = 2), pressure relaxation every 1.2 ps, SHAKE constraints, non-bonded interaction cut-off of 10 Å, and 2 fs integration time step. All other MD settings were kept the same as in one of our previous studies [50]. All MD trajectories were analyzed using AmberTools20 (http://ambermd.org, accessed on 9 June 2021). *N*-Glycan binding conformations were clustered into three clusters using the K-means clustering approach implemented in *cpptraj*.

The binding energy of both glycans in 1–3 and 1–6 binding modes was calculated using the MM/GBSA approach. A total of 1000 snapshots were extracted every 10 ps of a 100 ns MD simulation and used for the binding energy calculation with the MM/GBSA approach. In MM/GBSA, all explicit solvent water molecules and ions were stripped off, and solvation energy was calculated using the GB^HCT^ generalized-born model (igb = 1 in Amber) with set mbondi2 radii. The GB^HCT^ model outperformed the others in protein–glycan complexes [51]. We equate binding enthalpies to binding energies, assuming that entropic contributions do not play a major role in predicting the relative order of binding for the same system [52,53]. The salt concentration was 0.15 M. The surface tension and nonpolar solvation free-energy correction terms were set to 0.005 kcal·mol^−1^ and 0.0, respectively, for the solvent-accessible surface area (SASA) calculation. Exterior dielectric constants were set to 80. Other parameters were left as their default values in Amber20. The standard deviation of energy components from 1000 frames is plotted as the error bars (±SD) in the binding energy plot.

## Figures and Tables

**Figure 1 molecules-26-05176-f001:**
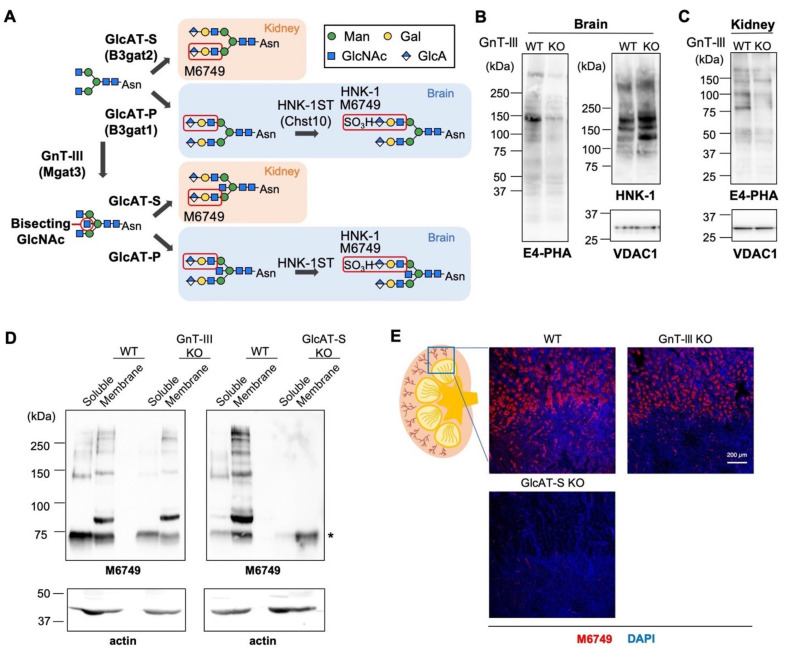
Unaltered expression of non-sulfated HNK-1 glycan in GnT-III KO mouse kidney. (**A**) Schematic diagram of HNK-1 biosynthesis. GnT-III catalyzes the addition of bisecting GlcNAc in *N*-glycan. GlcAT-P transfers GlcA, and HNK-1ST subsequently sulfates GlcA to produce HNK-1 epitope. In mouse kidney, GlcAT-S is expressed and synthesizes non-sulfated HNK-1 epitope that is recognized by M6749 mAb. (**B**) Proteins in brain membranes from adult WT or GnT-III KO mice were blotted with E4-PHA lectin, HNK-1 mAb, and anti-VDAC1 Ab. (**C**) Proteins in kidney membranes from adult WT or GnT-III KO mice were blotted with E4-PHA lectin and anti-VDAC1 Ab. (**D**) Soluble and membrane proteins from adult WT, GnT-III KO, or GlcAT-S KO mouse kidneys were blotted with M6749 mAb and anti-actin Ab. The asterisk indicates mouse IgM that is reactive to the secondary antibody. (**E**) Sections from WT, GnT-III KO, or GlcAT-S KO mouse kidney were immunostained with M6749 mAb and DAPI. Bar: 200 μm.

**Figure 2 molecules-26-05176-f002:**
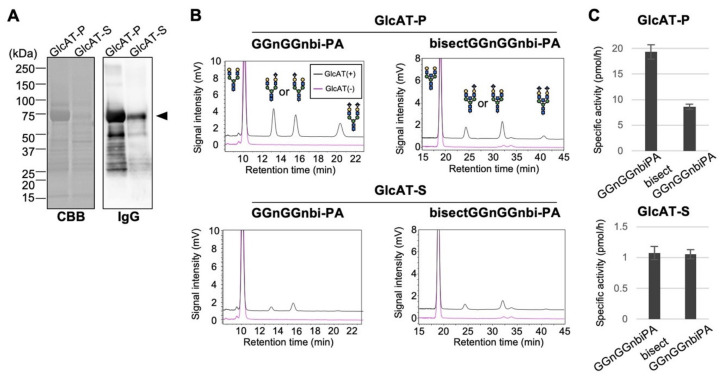
In vitro activity assay of GlcAT-P and GlcAT-S. (**A**) ProteinA-tagged GlcAT-P and GlcAT-S, which were expressed in COS7 cells and purified from the culture media, were subjected to CBB staining and Western blotting with HRP-conjugated IgG. (**B**) Purified GlcAT-P or GlcAT-S was incubated with the PA-labeled non-bisected (GGnGGnbi-PA) or the bisected (bisectGGnGGnbi-PA) acceptor glycan. The reaction mixtures were analyzed by reverse-phase HPLC. Amounts of the product having two GlcA residues eluted most slowly are smaller than the other two products, as in the case for other glycosyltransferases acting on *N*-glycan terminals [20]. (**C**) The peak areas in (**B**) were quantified, and the specific activity of GlcAT-P and GlcAT-S was calculated (*n* = 3). The graphs show means ± SD.

**Figure 3 molecules-26-05176-f003:**
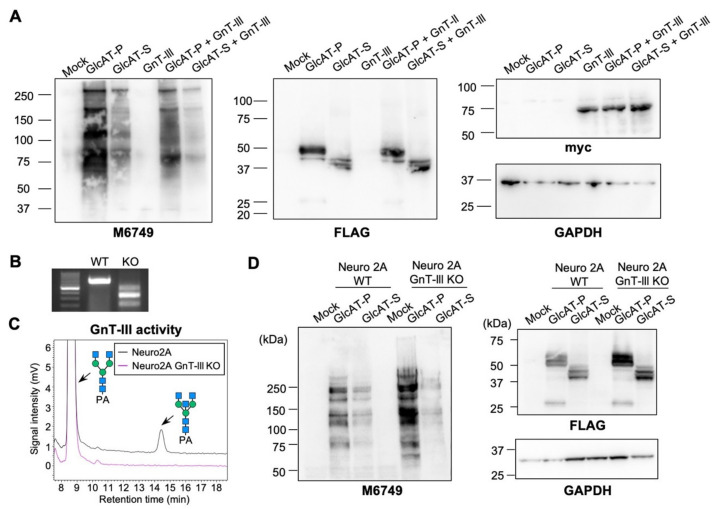
Effects of GnT-III overexpression or deletion on the intracellular activity of GlcAT-P and -S in cultured cells. (**A**) HEK293 cells were co-transfected with an expression plasmid for 3xFLAG-GlcAT-P or 3xFLAG-GlcAT-S, or an empty vector (Mock) with an expression plasmid for GnT-III or an empty vector. Cellular proteins were blotted with M6749 mAb, anti-FLAG M2 mAb, anti-Myc Ab, and anti-GAPDH Ab. (**B**) Genotyping PCR for the *Mgat3* gene using genomic DNA from WT and GnT-III KO Neuro2A cells. (**C**) GnT-III activity of WT and GnT-III KO Neuro2A cells was measured. Cell lysates were incubated with the acceptor substrate GnGnbi-PA, and the reaction mixtures were analyzed by reverse-phase HPLC. (**D**) WT and GnT-III KO Neuro2A cells were transfected with an expression plasmid for 3xFLAG-GlcAT-P or 3xFLAG-GlcAT-S, or an empty vector (Mock). Cellular proteins were blotted with M6749 mAb, anti-FLAG M2 mAb, and anti-GAPDH Ab.

**Figure 4 molecules-26-05176-f004:**
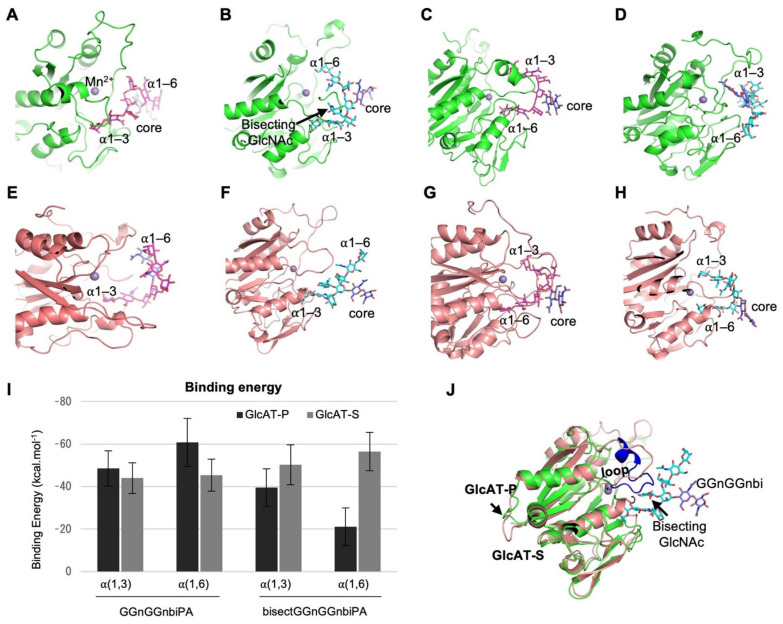
MD simulation of the binding of GlcATs and the *N*-glycans with or without bisecting GlcNAc. (**A**–**H**) Binding modes of non-bisected (GGnGGnbi, magenta) and bisected (bisectGGnGGnbi, cyan) biantennary *N*-glycans in GlcAT-P (green) and GlcAT-S (pink). The chitobiose core is colored in blue. These binding modes have been taken from the most populated conformation representative structure during 100 ns MD simulation. (**I**) The binding energy of bisectGGnGGnbi and bisectGGnGGnbi binding to GlcAT-P and GlcAT-S in kcal·mol^−1^. α(1,3) and α(1,6) indicate that this particular branch of *N*-glycan was kept in the reactive site, whereas the other branch was outside it. (**J**) Structures of GlcAT-P (green) and GlcAT-S (pink) superimposed on each other. The R142–I155 loop in blue occupies the position of bisecting GlcNAc in GlcAT-P.

## Data Availability

All the data are included in this manuscript.

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
