# Peer review of "Tissue-Specific Regulation of HNK-1 Biosynthesis by Bisecting GlcNAc"

_molecules, 2021, doi:10.3390/molecules26175176_

Round 1
Reviewer 1 Report
The authors continued their study on the production of the glycan HNK-1 and provided evidence that the enzyme GlcAT-S in the kidney is not affected as GlcAT-P is in the brain. The molecular background of this finding was then also supported by molecular modeling. This is a valuable output.
However, some points need clarification.
For example, it was confirmed that bisecting GlcNAc (formed by the GnT- III enzyme) decreased the amount of products formed by purified GlcAT-P (Fig. 2B top panel), and it was shown that purified GlcAT-S produced similar amounts of products, independent of bisecting GlcNac. However, the latter enzyme produced a lower amount of products and one type of product was absent. This is not adequately addressed.
The conclusion of the next experiment (overexpression of GlcAT enzymes in HEK293 cells) is unclear-this subsection leads to the statement that overexpression of GnT-III at high levels suppresses both GlcAT-P and GlcAT-S in cells by unknown mechanisms. It is unclear why a possible effect of bisecting GlcNAc is not discussed. Figure 1A does not clearly illustrate the corresponding sentence, which refers to it: "In contrast, GlcAT-S, but not GlcAT-P and 84 HNK-1ST, was shown to be selectively expressed in mouse kidney and to biosynthesize non-sulfated HNK-1 epitope in kidney (Figure 1A).
The electroforeograms of GlcATs (Figure 2A) show multiple bands. This needs to be commented on. The legend of Fig. 2C states "specific activity ... was calculated," but the y-axis title is missing from the corresponding graph.
The reader may want to know the degree of sequence identity of GlcAT-P and GlcAT-S ("highly homologous"). Is there a specific difference in their primary structures that leads to a different folding and, hence, a different affinity for bisecting GlcNAc?
The meaning of "bisecting GlcNAc" should be clarified at the beginning of the Introduction and in the Abstract.
Author Response
Comment 1
The authors continued their study on the production of the glycan HNK-1 and provided evidence that the enzyme GlcAT-S in the kidney is not affected as GlcAT-P is in the brain. The molecular background of this finding was then also supported by molecular modeling. This is a valuable output.
Response
Thank you very much for your positive comments that our paper has a valuable output. We have revised the paper according to the reviewers’ comments and hope that the revised version is now acceptable.
Comment 2
However, some points need clarification.
For example, it was confirmed that bisecting GlcNAc (formed by the GnT- III enzyme) decreased the amount of products formed by purified GlcAT-P (Fig. 2B top panel), and it was shown that purified GlcAT-S produced similar amounts of products, independent of bisecting GlcNac. However, the latter enzyme produced a lower amount of products and one type of product was absent. This is not adequately addressed.
Response
As the reviewer pointed out, GlcAT-S showed the lower amounts of the products than GlcAT-P in Fig. 2B. This is partially because the concentration of GlcAT-S is lower than GlcAT-P (Fig. 2A) and also because GlcAT-P always shows higher specific activity in in vitro assay than GlcAT-S (Kakuda et al., Glycobiology, 2005) (Kakuda et al., Prot. Expr. Purif., 2004). We have added the following sentence in the Results, “Overall activity of GlcAT-P was higher than GlcAT-S, consistent with the previous reports [31, 32]” (line 156-157).
The reason why one product (most right in HPLC) is absent in GlcAT-S assay is that the most right product has two GlcA residues in one molecule and produced after the other two types of products were formed. Therefore, in early phase of the reaction or in the case of weak activity of an enzyme, the peak is always smaller than the others or invisible. The peak of this most right product having two GlcA residues was also small in the case of GlcAT-P. Furthermore, this phenomenon is usually observed for many sialyltransferases and fucosyltransferases acting on N-glycan terminals (Nakano et al., Mol. Cell. Proteomics, 2019). To address this point, we have added the following sentence in the legend, “Amounts of the product having two GlcA residues eluted most slowly are smaller than the other two products, as in the case for other glycosyltransferases acting on N-glycan terminals [20]” (line 166-168).
Comment 3
The conclusion of the next experiment (overexpression of GlcAT enzymes in HEK293 cells) is unclear-this subsection leads to the statement that overexpression of GnT-III at high levels suppresses both GlcAT-P and GlcAT-S in cells by unknown mechanisms. It is unclear why a possible effect of bisecting GlcNAc is not discussed.
Response
According to the reviewer’s suggestion, we have added discussion regarding some possible effects of bisecting GlcNAc on GlcAT-P and GlcAT-S. As GlcAT-P and GlcAT-S are both N-glycosylated proteins, overexpression of GnT-III in cells probably caused very high expression of bisecting GlcNAc on these enzymes as well as on their substrate glycoproteins, which could in turn change some properties of GlcATs. Based on these points, we have added the following sentence, “Although the presence of bisecting GlcNAc suppressed the activity of GlcAT-P but not that of GlcAT-S in the enzyme assay (Figure 2), overexpression of GnT-III in cells might affect activity or localization of these enzymes as both GlcAT-P and -S are N-glycosylated” (line 193-195).
Comment 4
Figure 1A does not clearly illustrate the corresponding sentence, which refers to it: "In contrast, GlcAT-S, but not GlcAT-P and 84 HNK-1ST, was shown to be selectively expressed in mouse kidney and to biosynthesize non-sulfated HNK-1 epitope in kidney (Figure 1A).
Response
We have added “kidney” and “brain” in Figure 1A so as to illustrate the sentence more clearly. Thank you very much.
Comment 5
The electroforeograms of GlcATs (Figure 2A) show multiple bands. This needs to be commented on. The legend of Fig. 2C states "specific activity ... was calculated," but the y-axis title is missing from the corresponding graph.
Response
As pointed out, GlcATs show multiple bands in Fig. 2A. These are likely degradation products due to the high expression levels of the full enzymes. To indicate the positions of full length proteinA-GlcATs more clearly, we have added an arrowhead in Fig. 2A. Also, we have added the explanation of y-axis in Fig. 2C. Thank you very much for pointing it out.
Comment 6
The reader may want to know the degree of sequence identity of GlcAT-P and GlcAT-S ("highly homologous"). Is there a specific difference in their primary structures that leads to a different folding and, hence, a different affinity for bisecting GlcNAc?
Response
Thank you very much for your suggestion. We have added the identity (49.5%) of these two enzymes in Results (line 143-144).
Our previous crystal studies showed that these two enzymes have overall conserved structures but they also showed a few differences, which could lead to the different specificities of these enzymes. In addition to the different tolerances for bisecting GlcNAc shown in this study, we also reported that GlcAT-P showed very strict specificity toward LacNAc (Galbeta1-4GlcNAc) while GlcAT-S can accept both LacNAc and Lacto-N-biose (Galbeta1-3GlcNAc), which arises from the replacement of the two key amino acid residues in the acceptor binding site (Phe245 and Val320). Regarding the different specificities for bisecting GlcNAc, as shown in Fig. 4J, the position of R142–I155 loop in GlcAT-P is quite different from GlcAT-S. This loop is possibly in close proximity to bisecting GlcNAc and could be a reason for the weak affinity of GlcAT-P for bisecting GlcNAc. Accordingly, we have added a paragraph in Discussion, “Although the overall structures of GlcAT-P and GlcAT-S are highly conserved [33, 34], a few structural differences could lead to different specificities of these two enzymes. As shown in Figure 4J, the R142–I155 loop in GlcAT-P could cause a steric clash with bisecting GlcNAc. Furthermore, our previous study revealed that the replacement of the key residues (Trp234 and Ala309 in GlcAT-S, corresponding to Phe245 and Val320 in GlcAT-P) in acceptor sugar binding sites leads to the strict specificity of GlcAT-P toward LacNAc (Galb1-4GlcNAc) and tolerance of GlcAT-S for both LacNAc and Lacto-N-biose (Galb1-3GlcNAc) [34]” (line 260-267).
Comment 7
The meaning of "bisecting GlcNAc" should be clarified at the beginning of the Introduction and in the Abstract.
Response
According to the reviewer’s suggestion, we have modified the explanation of bisecting GlcNAc in Abstract, “bisecting GlcNAc, a branching sugar in N-glycan,” (line 16) and in Introduction, “Bisecting GlcNAc, the central GlcNAc branch linked to b-mannose synthesized by GnT-III (also designated as MGAT3) [9] (Figure 1A)” (line 51).
Reviewer 2 Report
The paper talks about two glycosyltransferase, GlcA-P and GlcA-S, regulated differently by the presence of the unique bisecting GlcNAc structure. The difference is reflected in tissue specificity and enzyme activity. Overall, the paper contains interesting information for the tissue-specific regulation of HNK-1 epitope in brain and kidney.
I have no question about the major structure of the paper. There is only one major concern about the GlcA-P and GlcA-S activity. Although Figure 2 showed GlcA-P activity is reduced when the bisecting GlcNAc structure exists, the HPLC profile still indicated that GlcA-P has much better activity than GlcA-S, even on the case of bisecting GlcNAc (Figure 2B). GlcA-S had no different activities toward non-bisecting GlcNAc and bisecting GlcNAc. However, both cases had relatively low activities (Figure 2C). The same observation has been observed in Figure 3A, 2nd and 3rd lines. The exogeneous expressed GlcA-P produced higher level of HNK-1 epitope and GlcA-S had much lower activity. We cannot understand whether the situation is derived from different enzyme concentrations. If the same protein concentrations were used, GlcA-P is generally active than GlcA-S.
The effect should be considered in the MD simulation (Figure 4). MD similarly indicated bisecting GlcNAc is not a good binder for GlcA-P, but a good one for GlcA-S. The conclusion got support from binding energy (Figure 4I). The paper used the evaluation to confirm the inhibition of bisecting GlcNAc on GlcA-P. However, if GlcA-P is actually a good enzyme for either non-bisecting and bisecting GlcNAc (just lower/reduced activity for bisecting GlcNAc) and GlcA-S is a weaker enzyme for both carbohydrate structures (just not sensitive to bisecting GlcNAc), then the explanation should re-write to make it clearer.
Minor:
- Figure 3: GAPDH for reference? The intensities are not equal.
- Figure 3D: GnT-III KO cells had much higher expression of GlcA-P (and GlcA-S). Any explanation? Are GlcA-P and GlcA-S endogenously expressed in Neuro 2A cells.
Author Response
Comment 1
The paper talks about two glycosyltransferase, GlcA-P and GlcA-S, regulated differently by the presence of the unique bisecting GlcNAc structure. The difference is reflected in tissue specificity and enzyme activity. Overall, the paper contains interesting information for the tissue-specific regulation of HNK-1 epitope in brain and kidney.
Response
Thank you very much for your positive comments on our manuscript. We hope that the revised version is now acceptable.
Comment 2
I have no question about the major structure of the paper. There is only one major concern about the GlcA-P and GlcA-S activity. Although Figure 2 showed GlcA-P activity is reduced when the bisecting GlcNAc structure exists, the HPLC profile still indicated that GlcA-P has much better activity than GlcA-S, even on the case of bisecting GlcNAc (Figure 2B). GlcA-S had no different activities toward non-bisecting GlcNAc and bisecting GlcNAc. However, both cases had relatively low activities (Figure 2C). The same observation has been observed in Figure 3A, 2nd and 3rd lines. The exogeneous expressed GlcA-P produced higher level of HNK-1 epitope and GlcA-S had much lower activity. We cannot understand whether the situation is derived from different enzyme concentrations. If the same protein concentrations were used, GlcA-P is generally active than GlcA-S.
The effect should be considered in the MD simulation (Figure 4). MD similarly indicated bisecting GlcNAc is not a good binder for GlcA-P, but a good one for GlcA-S. The conclusion got support from binding energy (Figure 4I). The paper used the evaluation to confirm the inhibition of bisecting GlcNAc on GlcA-P. However, if GlcA-P is actually a good enzyme for either non-bisecting and bisecting GlcNAc (just lower/reduced activity for bisecting GlcNAc) and GlcA-S is a weaker enzyme for both carbohydrate structures (just not sensitive to bisecting GlcNAc), then the explanation should re-write to make it clearer.
Response
As the reviewer pointed out, our data showed that GlcAT-P produced higher amounts of products in both enzyme assays (Fig. 2) and cell experiments (Fig. 3). This is partially because of the enzyme levels but mainly because GlcAT-P basically has higher activity than GlcAT-S in various enzyme assays. We previously showed that proteinA-tagged GlcAT-P purified from mammalian cells (Kakuda et al., Glycobiology, 2005) and FLAG-tagged GlcAT-P purified from E.coli (Kakuda et al., Prot. Expr. Purif., 2004) both showed higher specific activity than corresponding GlcAT-S. To address this point, we have added the following sentence in the Results, “Overall activity of GlcAT-P was higher than GlcAT-S, consistent with the previous reports [31, 32]” (line 156-157).
Even though the activity of GlcAT-S is totally lower than GlcAT-P, we reason that the different specificities toward bisected glycans shown in this study have significant effects on expression of HNK-1 in brain and kidney. In kidney, only GlcAT-S is expressed and knocking out GlcAT-S completely lost HNK-1 expression (Fig. 1D, E). Similarly, GlcAT-P knockout resulted in almost complete disappearance of HNK-1 in brain (Yamamoto et al., J. Biol. Chem., 2002) (Morita et al., J. Biol. Chem., 2009). Therefore, HNK-1 expression in brain and kidney almost exclusively depends on activity and specificity of GlcAT-P and GlcAT-S, respectively. Based on these points, we have modified the first paragraph in Discussion as follows, “Although the specific activity of GlcAT-P is totally higher than GlcAT-S [31, 32], HNK-1 expression in brain and kidney exclusively depends on GlcAT-P and GlcAT-S, respectively. Therefore, our findings suggest that the tissue-specific expression of the two isozymes (GlcAT-P and -S) with distinct tolerances for bisecting GlcNAc allows the tissue-specific regulation of HNK-1 synthesis by bisecting GlcNAc” (line 253-255).
Minor:
Comment 3
Figure 3: GAPDH for reference? The intensities are not equal.
Response
We have re-blotted the same samples and changed the GAPDH panel in Fig. 3A. Although we also reblotted the samples for Fig. 3D, we could not obtain clearer data. If clearer data are still needed, we are ready to carry out the same experiments again in the future.
Comment 4
Figure 3D: GnT-III KO cells had much higher expression of GlcA-P (and GlcA-S). Any explanation? Are GlcA-P and GlcA-S endogenously expressed in Neuro 2A cells.
Response
Regarding the high expression of GlcAT in GnT-III KO cells, we observed a similar higher expression of the exogenous protein in KO cells in our another project (using mouse Enpep cDNA). Therefore, we now think that this KO clone somehow has high efficiency of plasmid transfection for an unknown reason, and it is unlikely that protein expression of GlcATs is upregulated in KO cells.
GlcAT-P and GlcAT-S are not expressed endogenously in Neuro2A cells. We have added the following sentence, “As Neuro2A cells do not express GlcATs endogenously, exogenous expression of GlcATs resulted in expression of M6749-reactive HNK-1 glycans on various proteins (Figure 3D)” (line 193-195).
Round 2
Reviewer 2 Report
I still raised another question regarding MD simulation (Figure 4). MD similarly indicated bisecting GlcNAc is not a good binder for GlcA-P, but still a good one for GlcA-S, at least from the team of the binding energy (Figure 4I). That creates a misleading of GlcA-S activity is higher than GlcA-P. However, it might not be the case, in which GlcA-P is still more active than GlcA-S in bisecting GlcNAc. The authors need clarify this point either in discussion or result session.
Author Response
Thank you very much for your comment. We have added this sentence in the Discussion, “Although the MD simulation showed that a bisected glycan is not a good binder for GlcAT-P and a good binder for GlcAT-S, our enzyme assays showed that GlcAT-P has higher activity than GlcAT-S even for bisected glycans. The reason for this apparent discrepancy cannot be explained by our MD simulation at present” (line 257-260).